# Learning Blood Oxygen from Respiration Signals

## Abstract

Monitoring blood oxygen is critical in a variety of medical conditions. For almost a century, pulse oximetry has been the only non-invasive method for measuring blood oxygen. While highly useful, pulse oximetry has important limitations. It requires wearable sensors, which can be cumbersome for older patients. It is also known to be biased when used for dark-skinned subjects. In this paper, we demonstrate, for the first time, the feasibility of predicting oxygen saturation from breathing. By eliminating the dependency on oximetry, we eliminate bias against skin color. Further, since breathing can be monitored without body contact by analyzing the radio signal in the environment, we show that oxygen too can be monitored without any wearable devices. We introduce a new approach for leveraging auxiliary variables via a switcher-based multi-headed neural network model. Empirical results show that our model achieves good accuracy on multiple medical datasets.

## 1 Introduction

Oxygen saturation refers to the amount of oxygen in the blood – that is the fraction of oxygen-saturated hemoglobin relative to the total blood hemoglobin. Normal oxygen levels range from 94% to 100%. Lower oxygen can be dangerous, and if severe, leads to brain and lung failure (Taylor et al., 1951; Díaz-Regañón et al., 2002; Lapinsky et al., 1999). Oxygen Monitoring is important for patients suffering from lung diseases such as COVID-19, pulmonary embolism (PE), and chronic obstructive pulmonary disease (COPD) who are susceptible to reduced blood oxygen (NIH, 2020; Nordenholz et al., 2011). It is also recommended in old people since lung functions deteriorate with age putting the elders at a high risk of low oxygen (NLM, 2020).

Today, the only non-invasive approach for measuring oxygen saturation (SpO2) is pulse oximetry. It works by shining light on one's figure (or other body parts). Since oxygen-saturated hemoglobin has a bright red color, oxygen saturation can be measured by estimating the absorbance of the red-colored waves relative to other colors. While pulse oximetry is a very useful technology, it can be unsuitable for some of the most vulnerable patients. Oximetry requires the patient to wear a finger clip or other sensors, which can be cumbersome and hard to remember for older patients with dementia or cognition problems. Further, since it relies on measuring light absorbance, it is affected by skin color and tend to overestimate blood oxygen in dark-skinned subjects (Feiner et al., 2007).

In this paper, we introduce a new approach for monitoring blood oxygen that could be more appropriate for older patients or those with dark skin color. We propose to learn oxygen saturation from respiration signals. Recent advances in wireless sensing have demonstrated the feasibility of obtaining accurate measurements of the respiration signal from analyzing the radio waves that bounce off people's bodies, without any physical contact (Adib et al., 2015; Yue et al., 2018). Hence, if oxygen saturation is predictable from breathing, we can sense one's oxygen in a contactless way without any wearable sensors. Further, unlike pulse oximetry, such oxygen sensing method would be oblivious to skin color.

We model the problem as sequence-to-sequence learning and introduce a neural network that maps respiration signals to the corresponding oxygen saturation. A key innovation in our model is a new design for leveraging auxiliary variables. The standard approach in the literature for leveraging auxiliary variables would either provide those variables to the model as inputs or try to predict them as auxiliary tasks. We demonstrate empirically that for our problem neither of these approaches is beneficial. We show that in some cases, the gradient of the model can take vastly different and even divergent values given the auxiliary variable. In such a scenario, it is better to learn different

models for different values of the variable. This motivates us to propose an alternative design for incorporating auxiliary information. We introduce a multi-headed model where the value of the auxiliary variable is used to gate the learning through the appropriate head.

We evaluate our model on multiple medical datasets, and compare it with various baselines. Our empirical results show that the average absolute error in predicting oxygen saturation on the tested medical data is 1.6%, which is comparable to the accuracy of consumer pulse oximeters (whose average error ranges from 0.4% to 3.5% (Lipnick et al., 2016)). Further, by testing the model on data that include the RF modality, we show that the model generalizes to breathing signals extracted from radio waves, opening the door for continuous contactless monitoring of blood oxygen. We also use longitudinal data from one COVID-19 patient as a qualitative case study, showing that the predicted oxygen values are consistent with the patient's recovery process.

In summary, this paper makes the following contributions:

- The paper is the first to predict blood oxygen from respiration signals. This provides a novel non-invasive way for monitoring oxygen saturation, and improves our understanding of the relationship between these two physiological signals.
- The paper introduces a new approach for leveraging auxiliary variables. We analyze the gradient of a vanilla model with respect to the variable; if the gradient is divergent or vastly different for different values of the variable, we use a multi-headed neural network gated by the variable.
- The paper is the first to demonstrate the feasibility of contactless oxygen prediction from radio signals. Such a design can facilitate in-home oxygen monitoring, particularly for older patients who may have problems remembering to wear and charge oxygen sensors.

## 2 RELATED WORK

**Predicting Oxygen Saturation.** Past work on oxygen prediction focuses on inferring future oxygen values from recent measurements. Several papers use auto-regressive models that take past SpO2 readings and predict the SpO2 in the near future (ElMoaqet et al., 2013; 2014; 2016). Others use an auto-regressive model to model the photoplethysmogram (PPG) signal, which is the raw data obtained from a pulse oximeter (Lee et al., 2011). Some past work predicts changes in SpO2 in the five minutes after adjusting ventilator setting (Ghazal et al., 2019). Our work differs from these methods in that it predicts oxygen saturation from breathing, a completely different physiological signal. Further, we show the feasibility of oxygen prediction without body contact by inferring oxygen from radio-based breathing estimates.

**Contactless Sensing with Radio Frequency Signals.** The past decade has seen a rapid growth in research on passive sensing using radio frequency (RF) signals. Early work has demonstrated the possibility of sensing one's respiration and heart rate using radio signals (Adib et al., 2015). Building on this work, researchers have found that by carefully analyzing the RF signals that bounce off the human body, they can monitor a variety of health metrics including sleep, respiration, heart rate, gait, falls, and even human emotions (Nguyen et al., 2016; Wang et al., 2017; Yue et al., 2018; Hsu et al., 2017; Wang et al., 2016; Tian et al., 2018; Zhao et al., 2016; Jiang et al., 2018). Our work adds a new piece to the global picture of predicting vital signals from radio waves.

**Learning with Auxiliary Variables.** Improving machine learning model by leveraging auxiliary variables is a well-known idea. Treating auxiliary variables as extra input features is a simple but effective method, and has been shown useful in computer vision (Forsyth & Ponce, 2002), health profiling (Wang et al., 2019), and traffic prediction (Liao et al., 2018). Auxiliary learning (Liu et al., 2019) provides an alternative approach for leveraging auxiliary variables by taking them as secondary supervisors. It has been shown beneficial in multiple domains including object detection (Mordan et al., 2018), sequence modelling (Collobert & Weston, 2008; Dong et al., 2015), reinforcement learning (Wilson et al., 2007), and visual odometry (Valada et al., 2018). Our work explores a third way of using auxiliary variables. We focus on partitioning the data space by those auxiliary variables based on analyzing how they affect the gradient of the primary task. We show that a wise partition facilitates model learning and leads to better results.

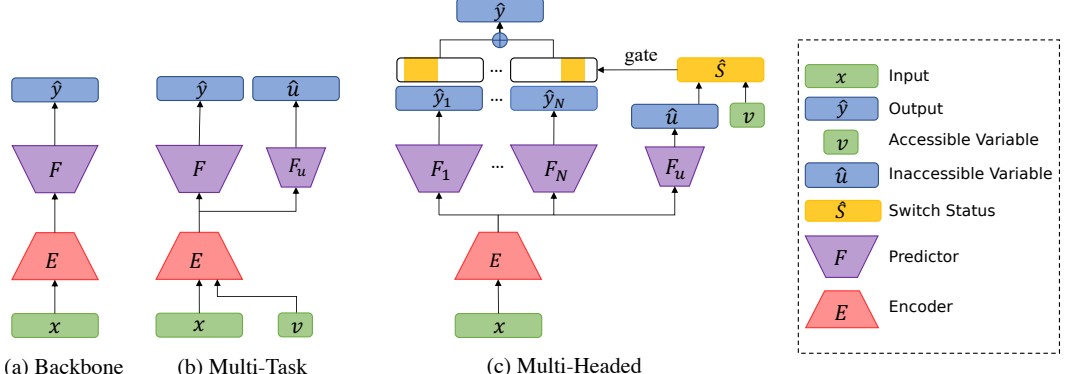

Figure 1: Illustrations of different models. (a) is the vanilla backbone model, which takes a sequence of breathing signal as input, and predicts the corresponding sequence of oxygen saturation. (b) is the multi-task model that has extra input $v$, and predicts extra output $\hat{u}$. (c) is the multi-headed model that generates multiple predictions under different status, and the final output is obtained by combining these predictions with the estimated status.

## 3 METHOD

### 3.1 PROBLEM FORMULATION

Our task is to predict oxygen saturation based on respiration signals[1]. Let the breathing signal be $x \in \mathbb{R}^{1 \times f_b T}$ and the corresponding oxygen saturation be $y \in \mathbb{R}^{1 \times f_o T}$, we aim at learning a sequence-to-sequence model that maps $x$ to $y$, i.e. $p(y^1, y^2, \cdots, y^{f_o T} | x^1, x^2, \cdots, x^{f_b T})$. Here, $T$ denotes the length of the segments (in seconds), $f_b$ and $f_o$ are the sampling frequencies of the breathing and oxygen data. In practice, we have high sampling rates for both respiration and oxygen saturation, i.e., $f_b = 10Hz$ and $f_o = 1Hz$. This allows us to capture instantaneous changes in these signals.

Moreover, we are particularly interested in monitoring oxygen saturation during sleep time. This is because blood oxygen typically drops during sleep, and supplemental oxygen is medically necessary if oxygen level drops to 88% or lower for 5 consecutive minutes during sleep (Becattini et al., 2018).

### 3.2 BACKBONE MODEL

As the relation between respiration $x$ and oxygen $y$ could be highly complicated, we use a deep neural network to map $x$ to $y$. We start with the backbone model shown in Figure 1(a), which consists of an encoder $E(\cdot; \theta_e)$ and a predictor $F(\cdot; \theta_f)$. The encoder is composed of a fully convolution network (FCN) followed by a bidirectional transformers (BERT) module (Devlin et al., 2018). The FCN extracts local features from the raw respiration signal, then the BERT module captures long-term dependencies based on those features. This design has two benefits. First, it can extract both local and global information from the respiration signal. Second, it works for variable length respiration signals. The predictor $F$ is composed of several De-convolution layers, which up-samples the extracted features to the same time resolution of oxygen saturation. Formally, we have $E : \mathbb{R}^{1 \times f_b T} \to \mathbb{R}^{n \times \alpha f_b T}$ and $F : \mathbb{R}^{n \times \alpha f_b T} \to \mathbb{R}^{1 \times f_o T}$ where $n$ is the size of the breathing feature and $\alpha$ is the down-sampling factor ($1/240$ in our model). The model is trained with a $L_1$ loss, i.e., $\mathcal{L}_o(F, E) = \frac{1}{T} \|\hat{y} - y\|_1$, where $\hat{y} = F(E(x))$ is the model prediction. Thanks to the elaborately designed model architecture, even with a simple $L_1$ loss, this model has achieved $1.66\%$ error (see Table. 1), which is a reasonably good result. More details on this model are available in Appendix A.

---

[1] The breathing signal is a time series that measures the chest displacement with the inhale-exhale motion. Please refer to Section B and Figure 7 for a detailed introduction and illustrative figures.

### 3.3 LEVERAGING AUXILIARY VARIABLES & THE MULTI-HEADED MODEL

In the medical domain, typically, there exist many physiological indices that could potentially help improve the model. From the viewpoint of machine learning, physiological indices can be viewed as auxiliary variables that could be beneficial to train the model. During testing, auxiliary variables can be categorized into two types: accessible and inaccessible. We denote accessible variables as $v$ and inaccessible variables as $u$. In the learning procedure, a common practice is to use accessible variable as extra inputs and use inaccessible variables for auxiliary tasks. Figure 1(b) illustrates such framework where $F_u$ denotes a predictor of the auxiliary task $u$. We call the design in Figure 1(b) the *Multi-Task* model. Despite its intuitive appeal, in practice, this model may not always lead to improvements. As shown in Table 1, the Multi-Task model even causes a performance drop on the SHHS dataset[2].

In our work, we investigate an alternative approach that uses auxiliary variables as switches. The idea is based on the hypothesis that some variable $h$ affects the target input-output relationship in a non-smooth way. For example, assuming $h$ is a binary variable. The mapping from input $x$ to output $y$ conditioned on variable $h$ equal to 0 is significantly different from the mapping when the variable is 1, i.e., $y(x|h = 0)$ and $y(x|h = 1)$ are two distinct functions. In this case, instead of learning function $y(x, h)$, we propose to use variable $h$ as a switch to partition the data space and learn the functions $y(x|h = 0)$ and $y(x|h = 1)$ separately. We realize this idea in our *Multi-Headed* model illustrated in Figure 1(c). We first define a switch function $S(v, u) : \mathcal{V} \times \mathcal{U} \to [N]$ where $v \in \mathcal{V}$ and $u \in \mathcal{U}$ are accessible/inaccessible variables and $N$ is the number of switch status. The model has $N$ heads $\{F_i\}_{i=1}^N$ to make the prediction $\hat{y}_i = F_i(E(x))$ for every switch status separately. It also has a predictor $F_u$ to infer $u$. During testing time, based on the accessible auxiliaries $v$ and estimated inaccessible auxiliaries $\hat{u}$, we evaluate the switch status $\hat{S} = S(v, \hat{u})$, and output the final prediction $\hat{y} = \hat{y}_{\hat{S}}$. In the case of oxygen prediction, $\hat{y}_i$ and the switch $\hat{S}$ are time series, we use $\hat{y}_i^t$ and $\hat{S}^t$ to denote their value at time step $t$. As shown in Figure 1(c), the final prediction is the gated combination of every head's output, $\hat{y}^t = \hat{y}_{\hat{S}^t}^t$. One might argue that the Multi-Task model can also learn such compositional function by implicitly learning each piece of the target function, and learning to assign the input to the target piece based on the value of the auxiliary variable. However, in practice, such incorrect model inductive bias introduces unnecessary difficulty in the learning procedure and typically leads to bad performance. In our experiments, we show, with a more preferable inductive bias, the Multi-Headed model outperforms the Multi-Task model with a clear gap.

### 3.4 IDENTIFYING GOOD SWITCHES VIA GRADIENT DIAGNOSIS

In the above description, we assume a good switch function exists. In this paragraph, we answer the critical question: which auxiliary variables should be used as switches? We introduce, gradient diagnosis, a simple but effective way to test the influence of variable $v$ on the task of learning function $y(x)$. We first train a vanilla model with no auxiliary variables, e.g., our backbone model. We then use the validation set to check the predictor $F(\cdot; \theta_f)$'s loss gradient $\nabla_{\theta_f} \mathcal{L}$. For each possible value $a \in \mathcal{V}$ of variable $v$, we compute the gradients averaged on data samples when $v$ equals $a$, i.e., $\nabla_{\theta_f} \mathcal{L}|_{v=a} = \mathbb{E}_{(x,y) \sim p(x,y|v=a)} \nabla_{\theta_f} \mathcal{L}(F(E(x)), y)$. Such gradient indicates the direction to improve the model when $v = a$. Intuitively, if gradients $\nabla_{\theta_f} \mathcal{L}|_{v=a}$ are similar for all values $a$, it means variable $v$ does not change the loss function drastically. So it is reasonable to use one model to handle all values of $v$. On the other hand, if the gradients are vastly different from each other, it is preferable to separate the predictor for different values of $v$.

## 4 EXPERIMENTS

We present the results of empirical evaluation on medical datasets and RF data. We evaluate the models using the average absolute error between the predicted oxygen saturation $\hat{y}$ and the ground-truth $y$. The error is first averaged for every night and each subject, i.e., $\frac{1}{T} \sum_{t=1}^T \|y^t - \hat{y}^t\|$. We then report the mean and standard deviation of per-night averaged errors of the dataset.

---

[2]More dataset details are provided in Section 4.1.

## 4.1 DATASETS

**Medical Datasets.** We leverage three publicly-available medical datasets: Sleep Heart Health Study (*SHHS*) (Zhang et al.), Multi-Ethnic Study of Atherosclerosis (*MESA*) (Chen et al., 2015), and Osteoporotic Fractures in Men Study (*MrOS*) (Blackwell et al., 2011). Each dataset contains the polysomnography (PSG) signals of subjects. The first two datasets have males and females, whereas the last one has male subjects. The PSG data includes full-night respiration signals and the corresponding oxygen saturation (SpO2) and sleep stage labels. The data has three types of respiration channel measured with: a nasal probe, a breathing belt on the chest, and a breathing belt on the abdomen. We denote it as *BB-respiration signals*. The datasets also contain subjects' bio-information such as gender, age, etc. SHHS, MESA, and MrOS contains 2651, 2056, and 1026 subjects, respectively. We randomly split the training and testing sets for SHHS, MESA, and MrOS as 1855/796, 1439/617, and 718/308, and keep the same splits in all experiments. We note that the subjects in these studies have an age range between 40 and 95, and suffer from a variety of diseases as indicated by the studies names. This allows for a wider range of oxygen variability beyond the typical range of healthy individuals.

**RF Dataset.** We collect a small PSG dataset that contains the standard PSG channels mentioned above as well as an RF-based inference of the respiration signal, i.e., *RF-respiration signals*. The dataset is collected in the sleep lab in the university and contains 31 nights of data from 14 individuals. All subjects in this study are healthy. Each subject is monitored using the standard PSG channels as in the medical datasets above (SpO2, respiration belts, sleep stages, etc.). Additionally, a radio device is installed in the room to collect RF signals, which are then processed to extract the subject's respiration in accordance with Adib et al. (2015). Therefore, in this dataset, each subject has both the BB-respiration signals and the RF-respiration signals.

## 4.2 MODELS

As we describe in Section 3, our models focus on predicting oxygen from breathing while leveraging auxiliary gender and sleep stage information. We evaluate the following models: (a) *Single-Headed*: the vanilla backbone model. (b) *Multi-Task*: the model that takes gender as an extra input and is trained with sleep stages as an auxiliary task. (c) *Multi-Headed*: the model that uses multiple heads to predict oxygen level for different switch status. (d) We also compare with an adaptation of the Projecting Conflicting Gradient *PCGrad-Adapted*. The original PCGrad was proposed for improved multi-task learning to deal with scenarios where the various tasks have divergent gradients (Yu et al., 2020). In that case, PCGrad projects the divergent gradients along consistent directions. In our context, the gradients of the tasks themselves – i.e., oxygen prediction and sleep stage prediction – do not diverge. Hence, simply applying PCGrad to our problem should be no different from the Multi-Task model. However, we can adapt PCGrad to be more suitable for our problem. In our case, the gradients of subgroups (i.e., subjects with different status) in the input data with respect to the main task can be divergent or significantly different. We can think of learning a mapping from breathing to oxygen for different subject status as different sub-tasks. Then, we can augment our backbone model with PCGrad applied to these sub-tasks. Specifically, to train the single predictor $F$, We compute 6 different losses based on the six subject status corresponding to our 6-headed model. For each subject status $i$, we have an oxygen prediction loss $\mathcal{L}_o^i$ for that status. Mathematically, $\mathcal{L}_o^i(F, E) = (\sum_{t:S^t=i} 1)^{-1} \sum_{t:S^t=i} |F(E(x_b))^t - y^t|$. We compute the predictor's gradients $\nabla_{\theta_f} \mathcal{L}_o^i$ for each loss and update the predictor by the gradient calibrated by PCGrad.

## 4.3 GRADIENT DIAGNOSIS RESULTS

Before evaluating the various models, we show that the gradient diagnosis method proposed in section 3.3 finds good "switches" for the oxygen prediction task. Among the huge set of physiological indices in the medical datasets, we pick the following variables as candidates: (a) Sleep stages: awake, light sleep (N1,N2), deep sleep (N3), rapid eye movement (REM); (b) Gender: male vs. female; (c) Asthma: having asthma or not; (d): Smoking: non-smoker, current smoker, former smoker; (e) Education: less than 15 years vs. greater than 15 years; (f): Height: lower than 160cm or higher than 160cm; (g) Aspirin: taking aspirin or not. Figure 2 visualizes the cosine similarity of the gradients computed on the medical dataset SHHS. We clearly see that the gradient directions are altered by the sleep stage, gender, asthma and smoking while less affected by the

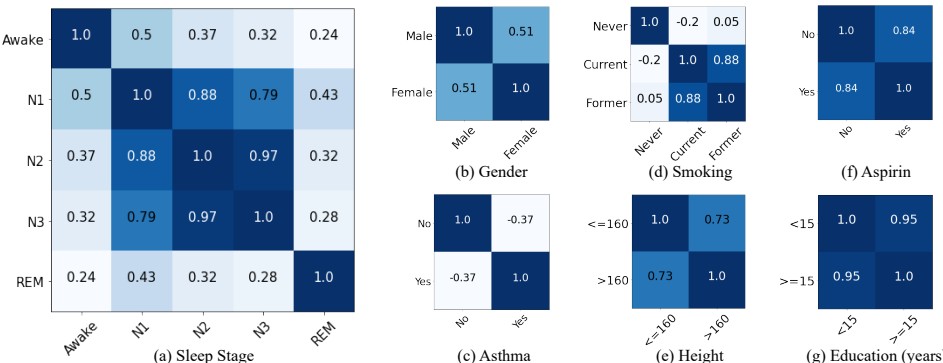

Figure 2: Gradient cosine similarities of different variables.

height, aspirin and education level. These result is quite interesting. For example, although it is hard to see why gender affects the oxygen-breathing function from the first glance, there are medical evidence that show gender matters. Due to differences between women's and men's respiratory system, women consume more oxygen when breathing compared to men (Ramonatxo, 2003), especially during exercise (Reybrouck & Fagard, 1999), and men's oxygen saturation is naturally higher than women (Ricart et al., 2008). As for sleep stages, it is a known fact that it influences the oxygen saturation. For example, pointed out by Choi et al. (2016), that oxygen saturation differs significantly during REM and Non-REM sleep. It is also understandable that asthma affects oxygenation since asthma causes difficulty in breathing. It is also supported by medical findings that smoking reduces oxygen saturation (Özdal et al., 2017). Since it is a long-term effect, no matter if a smoker quits or not, his/her oxygen pattern is already altered. This may explains why the gradient similarity is high between current smoker and former smoker.

Based on our gradient diagnosis, we pick sleep stage and gender as auxiliary variables for our task. Other variables are possible. However, we pick these two variables because gender is almost always accessible during testing; and while sleep stages are usually not available during testing, luckily past work shows that sleep stages can be inferred from respiration signals (Zhao et al., 2017). Hence, our model can estimate the sleep stages and use the estimates to do the switching. Further, from Figure 2(a), we can see the gradients for awake, REM and sleeping (N1,N2,N3) are very dissimilar from each other while the gradients are much more similar within the group of N1,N2,N3. So we design the switch function to consider N1,N2,N3 as one sleep stage. In total, we have six switch status due to the combinations of gender and sleep stages. Our model has six heads for each switch status.

### 4.4 EVALUATION ON MEDICAL DATASETS

**Quantitative Results.** Table 1 reports the absolute error of all models for the three medical datasets. The table reveals two main results. First, all baselines perform reasonably well and have average prediction errors between 1.58% and 1.7%. These errors are relatively low and comparable to the errors of consumer pulse oximeters whose average error is in the range 0.4-3.5% (Lipnick et al., 2016). It is important to keep in mind that all baselines are based on our backbone model, which already embodies our design for learning this new task. Thus, these results show for the first time that respiration signals have a predictive power for learning oxygen saturation.

Second, Table 1 also shows that the Multi-Headed model outperforms the other baselines on all datasets. This demonstrates that the Multi-Headed design is more effective at leveraging relevant auxiliary variables than the other methods.

**Qualitative Results.** Figure 3 visualizes the predicted oxygen saturation of the Multi-Headed model and the Multi-Task model on a male subject in the SHHS dataset. As the ground-truth oxygen saturation are integers, we rounded the predicted oxygen saturation into integers. The background color indicates different sleep stages. The 'light grey' and 'white' corresponds to 'REM' and 'Awake', respectively. For clarity, we use one color 'dark grey' for all three non-REM 'Sleep' stages, N1, N2, and N3. We observe that the Multi-Headed model consistently performs better than the Multi-Task model over the whole night. In non-REM 'Sleep', the ground-truth oxygen saturation is more stable.

Table 1: Prediction error (mean $\pm$ std) of different models for three medical datasets.

| Model | SHHS | MESA | MrOS | Overall |
|---|---|---|---|---|
| Single-Headed | $1.68 \pm 1.56$ | $1.59 \pm 1.58$ | $1.76 \pm 1.56$ | $1.66 \pm 1.53$ |
| Multi-Task | $1.69 \pm 1.57$ | $1.51 \pm 1.56$ | $1.71 \pm 1.57$ | $1.62 \pm 1.53$ |
| PCGrad-Adapted | $1.74 \pm 1.58$ | $1.62 \pm 1.57$ | $1.74 \pm 1.53$ | $1.71 \pm 1.54$ |
| Multi-Headed | $\mathbf{1.62 \pm 1.54}$ | $\mathbf{1.48 \pm 1.51}$ | $\mathbf{1.66 \pm 1.51}$ | $\mathbf{1.58 \pm 1.49}$ |

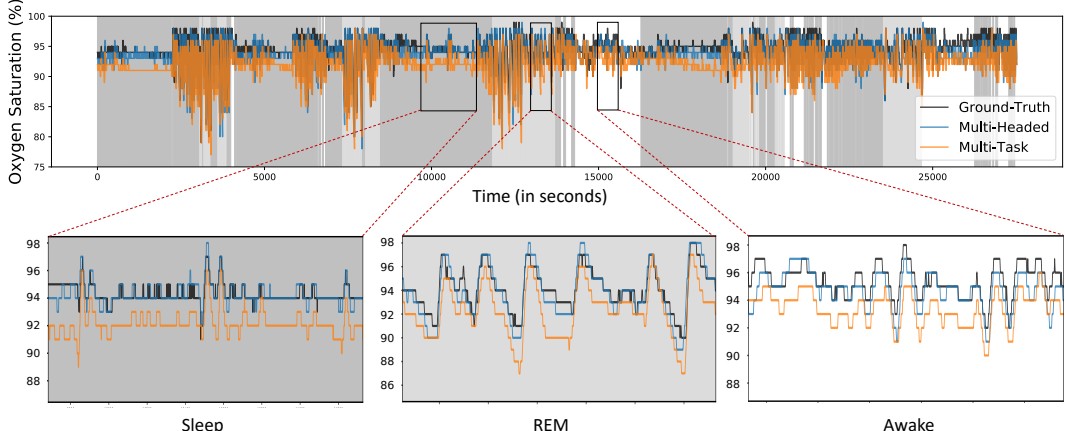

Figure 3: Visualization of the oxygen saturation predicted by the Multi-Task model and the Multi-Headed model on a male subject in SHHS dataset. The background color indicates sleep stages. The 'dark grey', 'light grey' and 'white' corresponds to 'Sleep' (N1, N2, N3), 'REM' and 'Awake'.

The Multi-Headed model predicts more accurate oxygen than the Multi-Task model. In the 'REM' stage, people usually have lower oxygen saturation with larger variance. The Multi-Headed model predicts the ups and downs and captures the changes in blood oxygen. In the 'Awake' stage, the Multi-Headed model predicts the more dramatic changes, while the Multi-Task model has a larger bias. This experiment demonstrates that the way we incorporate the sleep stages into the model improves the performance across different sleep stages.

**Error Analysis and Relation to Skin Color.** One important issue to keep in mind is that our model learns from ground-truth data measured using pulse oximeters (today oximetry is the only way for continuous noninvasive monitoring of blood oxygen). Hence, to understand the prediction errors and their potential causes, we plot in Figure 4 the distributions of the ground-truth SpO2 and the predicted SpO2 by the Multi-Headed model for different races, for all datasets. Interestingly, the ground-truth SpO2 from oximetry shows a clear discrepancy between black and white subjects. In particular, the ground-truth oxygen distributions show that black subjects have higher blood oxygen. This is compatible with past findings that pulse oximeter overestimates blood oxygen in dark-skinned subjects (Feiner et al., 2007). In contrast, the model prediction shows much more similar oxygen distributions for the two races. It is interesting that while the model learned from what seems to be biased ground-truth, the results show that the model is able to correct, to some extent, for this bias. Admittedly, these observations are inductive since there is no way to measure the exact errors in the ground-truth labels.

The distributions in Figure 4 also show that the model tends to miss the very high and very low oxygen values. This is expected given that only a very small percentage of the ground-truth data falls in the tails. One would expect that the performance on very high and very low oxygen values would improve if there is sufficient training data in those ranges.

## 4.5 EVALUATION ON RF DATASET

In this section, we evaluate whether our model, which is trained on respiration measured using a breathing belt, can generalize to predicting oxygen from respiration signals measured via radio waves. We directly apply a Multi-Headed model trained on the BB-respiration signals in medical

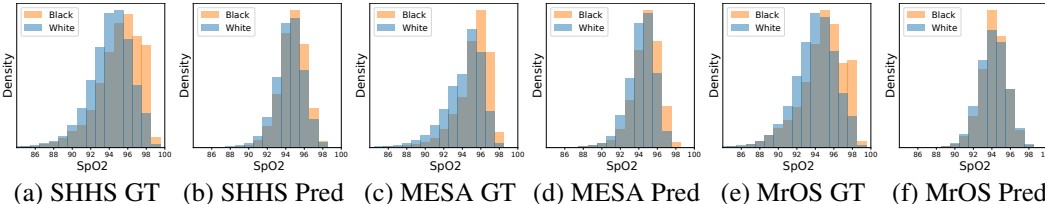

(a) SHHS GT    (b) SHHS Pred    (c) MESA GT    (d) MESA Pred    (e) MrOS GT    (f) MrOS Pred

Figure 4: Distributions of the ground-truth SpO2 readings ((a), (c), (e)), and the predicted SpO2 ((b), (d), (f)) by Multi-Headed model for the two races.

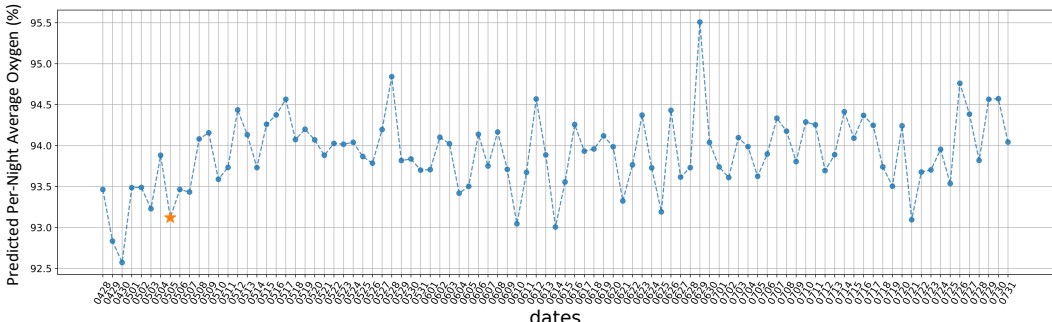

Figure 5: The predicted per-night averaged oxygen saturation for a COVID-19 patient from Apr 28 to July 31, 2020. The patient turned negative for COVID-19 on May 5, 2020 (marked by the star).

datasets, to the RF dataset. For comparison, we test the model on both RF-respiration signals and BB-respiration signals from the RF dataset. The results are shown in Table 2. We observe that the model works well on RF dataset with a small prediction error, no matter which breathing signal is used. The prediction error of using the RF-respiration signals (1.28%) is even slightly lower than using the BB-respiration signals (1.30%). This is because the noise in the respiration measured using RF signal is different from the noise in the breathing belt measurements. The noise in the RF-based measurements tend to increase with motion and hence we see that the model prediction error is higher for the aware stage. In contrast, the error in the breathing belt depends on the sleep posture and the positioning of the belt on the body. Also, comparing with the error from previous section, we find the model is performing better on the RF dataset than on the medical datasets, i.e., 1.30% vs. 1.62%. We believe the reason is that the RF dataset is collected with healthy individuals and hence has less complexity and is easier to predict. These results show the first demonstration of predicting oxygen saturation from radio signals without any body contact.

Table 2: Prediction error for the RF dataset using BB-respiration signals or RF-respiration signals.

| | BB-respiration signals | | | | RF-respiration signals | | |
|---|---|---|---|---|---|---|---|
| Awake | Sleep | REM | All | Awake | Sleep | REM | All |
| 1.37 | 1.30 | 1.27 | 1.30 | 1.65 | 1.24 | 1.35 | 1.28 |

### 4.6 APPLICATION TO MONITORING COVID-19 PATIENTS

Finally, we show a case study of a COVID-19 patient. An 88-year old male was monitored from April 28 till July 31, in home, with a contactless radio device. The subject tested COVID-19 positive prior to the monitoring, and was declared COVID-19 negative on May 5. The objective of the case study is to compare the results of a completely passive contactless monitoring approach to information collected from the medical doctor and the professional caregiver attending to the subject.

Figure 5 shows the average oxygen saturation predicted by the Multi-Headed model for each of the monitored nights. May 5 is highlighted with a star as the day on which the patient was declared COVID-19 negative. The results in the figure show a clear trend of increased oxygen level as the patient transitioned to becoming COVID-19 negative. They also show that improvement in blood oxygen did not stop on the day the patient became COVID-free; it rather continued to ramp up for almost two weeks after that date, and eventually reached a steady level. While there is no ground-

truth measurements of the oxygen for this completely passive case study, the results are consistent with the doctor notes about patient recovery and general expectations for a symptomatic COVID-19 patient who recovered without escalation of his COVID-19 symptoms.

## 5 DISCUSSION AND CONCLUDING REMARKS

This paper introduces a model for predicting oxygen saturation from breathing signals, and shows that it performs well on three medical datasets and an RF-based dataset. In this section, we discuss some of the clinical implications, properties, and limitations of such model.

**Data Diversity.** We note that the medical datasets used in our evaluation include many patients with diverse diseases and health conditions. In particular, the Multi-Ethnic Study of Atherosclerosis (MESA) dataset focuses on patients with Atherosclerosis, a disease in which cholesterol plaques build up in the walls of arteries, and the minimum oxygen saturation predicts disease prognosis (Gunnarsson et al., 2015). As shown in the results, our model has low error and good performance on this dataset. In fact, all three evaluation datasets (SHHS, MrOS and MESA) contain people who have various diseases including chronic bronchitis (323 subjects), cardiovascular diseases (1196 subjects), coronary heart disease (869 subjects), myocardial infarctions (508 subjects), diabetes (307 subjects), Alzheimer's (40 subjects), Parkinson's (69 subjects), and others. This means that our reported accuracy already takes into account unhealthy individuals. We note that many of these diseases, including diabetes and pulmonary and cardiovascular diseases, interact with oxygen. Our model performs well on this population, as shown in our evaluation results. Furthermore, all three datasets are focused on old people. In particular, the median age in MrOS is 80 years. Hence, 50% of the subjects are in their 80's and 90's. Such old subjects are natural beneficiaries from monitoring oxygen saturation since they are at a high risk of low oxygen (NLM, 2020).

**Relationship between breathing signals and Oxygen.** Oxygen affects breathing through the peripheral chemoreceptors, located in the carotid bodies (Plataki et al., 2013). There is a feedback loop that relates blood oxygen and ventilation, allowing changes in oxygen to cause changes in breathing. However, the details of the chemoreflex physiology are complex and not yet fully understood. Our approach is data-driven and demonstrates that custom neural networks can extract significant and useful information about oxygen level from the breathing signal. It is important to note that the model learns more than the relationship between breathing rate and oxygen level. In Section B, we show several examples where two subjects breath at the same rate, yet the model detects that their oxygen levels are quite different. Such empirical results show that the model not only leverages the breathing rate, but also other more sophisticated patterns in the detailed breathing signals.

**Limitations and Clinical Usefulness.** Oxygen saturation and breathing are physiologically related, but this does not mean that the model can discover all situations that lead to low oxygen. Even pulse oximetry does not always work. The literature reports that in the critically ill, oximetry does not reliably predict changes in oxygen saturation because the calibration of red-light absorbance is based on calculations made from healthy volunteers (Ralston et al., 1991). Oximetry also fails in subjects with COPD (Amalakanti & Pentakota, 2016), anemia (Severinghaus & Koh, 1990), sickle cell disease (Blaisdell et al., 2000), skin pigmentation (Bickler et al., 2005). Similarly, the model herein has its own limitations and cannot detect low oxygen if the breathing signal is not impacted.

Our objective is to introduce a method that is non-invasive, easy to use, non-contact, and can be useful in many cases, particularly for old people, who are naturally at high risk of reduced oxygen (NLM, 2020). This population is well represented in our evaluation datasets, on which the model is demonstrated to have good performance. Furthermore, old patients can have onset of dementia or Alzheimer's and cannot remember to regularly measure themselves using a pulse oximeter. Thus, they would benefit from a system that passively and continuously measures oxygen using radio signals, without asking the patient to remember to actively measure themselves.

The model can also provide a complementary oxygen monitoring system at home or in the hospital. As explained above pulse oximetry fails in various scenarios. One can add a radio device in the patient room to collect radio signals and use our system to augment oximetry. This does not add any overhead to the patient since they will not wear any additional sensor but rather be monitored in a passive contactless manner using radio signals. Further, since oximetry and our model use intrinsically different methods, they have independent errors, and hence can complement each other.

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

## A  MODEL ARCHITECTURE AND IMPLEMENTATION DETAILS

As we described in section 3, our vanilla model consists of an encoder and a predictor. The encoder has nine 1-D convolutional layers (Conv-Norm-RReLU) that shrink the temporal dimension of the features by 240 times. It then followed by several bi-directional multi-head self-attention layers (Devlin et al., 2018) to aggregate the temporal information at the bottleneck. We use 8 layers, 8 heads with hidden-size of 256, intermediate-size of 512 for self-attention, and the max position embeddings is 2400. The decoder contains 7 layers of 1-D de-convolutional layers (DeConv-Norm-RReLU). We also use a skip connection (Ronneberger et al., 2015) by concatenating the convolutional layers in the encoder to the de-convolutional layers in the predictor. Figure 6 illustrate the overall network architecture. We keep the encoder and predictor architectures the same for all models. All models are implemented using PyTorch. The number of parameters for the multi-headed model is 26,821,113 and the model size is 107.28MB. In the training process, we use the Adam optimizer with a learning rate of $2 \times 10^{-4}$, and train the model for 500 epochs. Due to the varying input length, we set the batch size for all the models to 1 (i.e., one night of breathing signals and the corresponding oxygen time series).

Because oxygen saturation is affected by sleep stages and sleep cycles, it exhibits non-local dependencies. To allows the model to capture these dependencies, we use the breathing signals from a night of sleep as one input sample. To process such a long signal at the input, we use a UNET structure with a bottleneck that shrinks by a factor of 240, and has a max position embedding of 2400. The breathing signal is measured at 10 fps (frames per second). Thus, the model can process $240 * 2400/10/3600 \approx 16$ hours of breathing data.

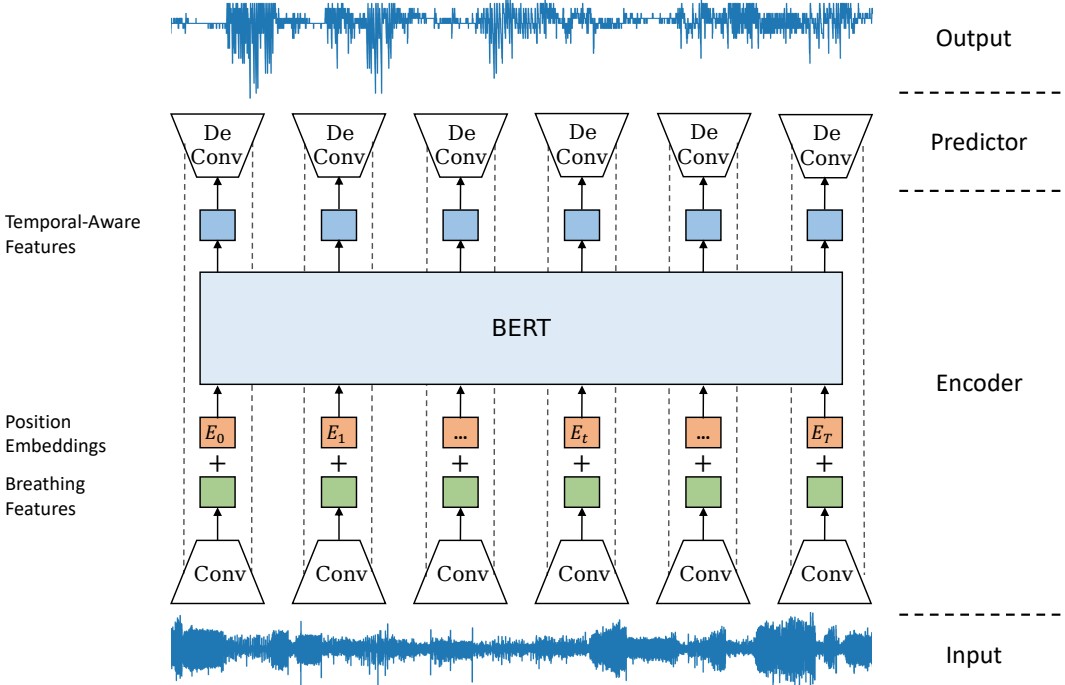

Figure 6: Network architecture for the vanilla model.

# B BREATHING SIGNALS

The input respiration signal we used in the paper is a one-dimensional time series with a frame rate of 10 fps (frames per second). It contains full information about the inhale and exhale motions. Figure 7 illustrates typical patterns of breathing signals with the corresponding oxygen saturation. Figure 7(a) shows a normal breathing pattern, which leads to constant oxygen saturation. Figure 7(b-d) are abnormal breathing signals.

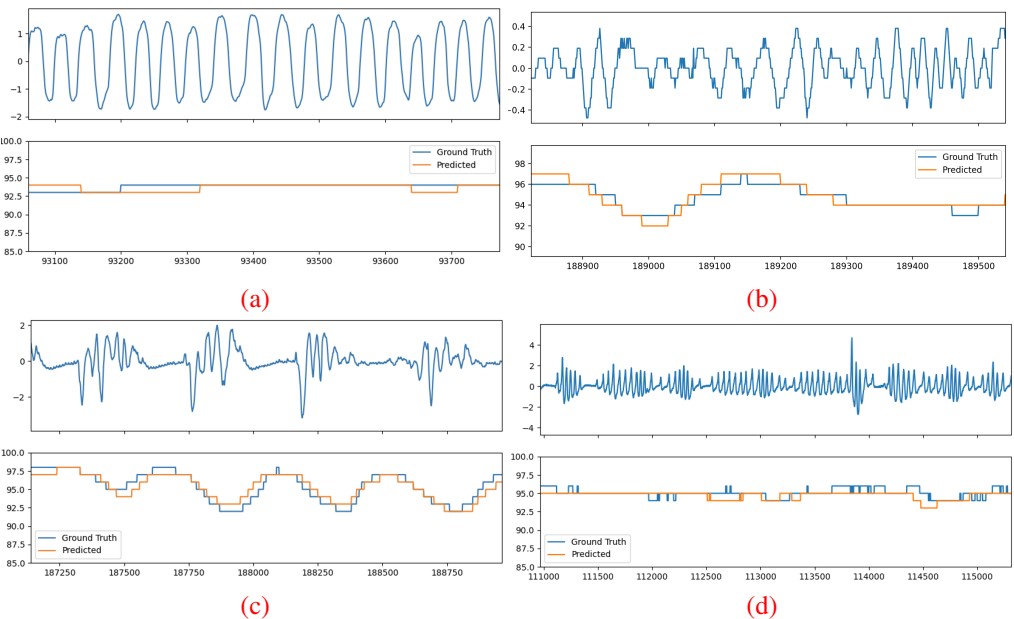

(a)          (b)

(c)          (d)

Figure 7: Visualization of breathing signals and the corresponding oxygen level. (a) is an example of normal breathing signals. (b-d) are examples of different abnormal breathing signals. In every figure, the first row shows the breathing signals while the second row shows the ground truth oxygen level (in blue) and the oxygen level predicted by our model (in orange).

## B.1 FULL BREATHING SIGNALS VS. BREATHING RATE

It is important to note that we use the full breathing signals instead of the breathing rate. Full breathing signals contain richer information about oxygen saturation than the breathing rate. Figure 8 shows several cases where two subjects have the same breathing rate but different oxygen saturation values. The figure also shows that our model is able to predict the correct oxygen levels and is not confused by the fact that the subjects have the same breathing rates.

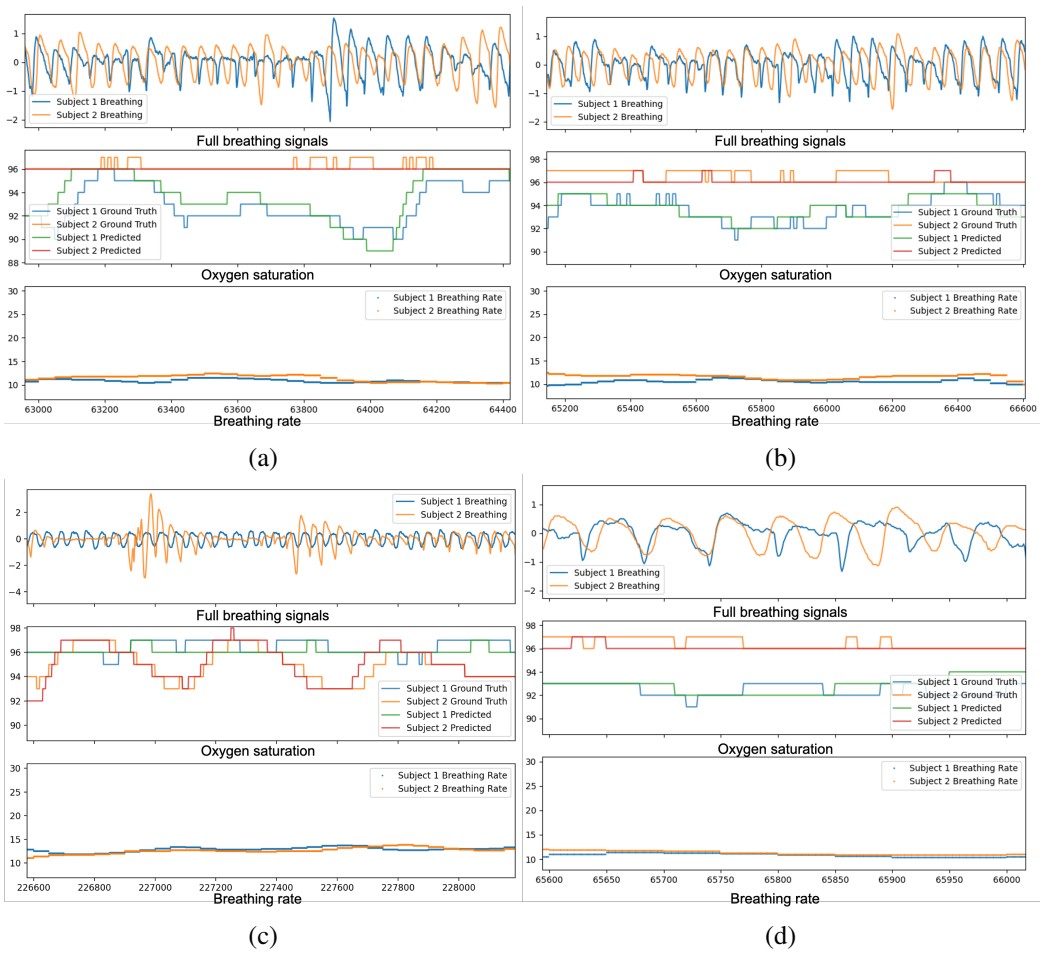

Figure 8: Visualization of full breathing signals having more information about oxygen level than breathing rates. Here are four examples of subjects having very similar breathing rate but different oxygen saturation. However, if we look at the full breathing signals, we can still differentiate them. Our model can capture the shape difference of full breathing signals and make the correct prediction.

## C    MORE VISUALIZATIONS

We include more visualizations of the breathing signals and the corresponding oxygen saturation predicted by the *Multi-Headed* model for the medical datasets: MESA (Figure 9 and Figure 10), MrOS (Figure 11 and Figure 12). In the plots, The background color indicates different sleep stages. The 'dark grey', 'light grey' and 'white' corresponds to sleep stages (N1, N2, N3), 'REM' and 'Awake', respectively.

### C.1    MESA

The examples in Figure 9 show our model's ability to capture the fluctuations of oxygen saturation. At the same time, the examples in Figure 10 shows our model accurately detects the region of low oxygen saturation, which highlights its usefulness in monitoring patients.

### C.2    MROS

Looking at the zoomed-in range (a) and (b) from Figure 11, we see that the model exhibits a larger error when the ground truth SpO2 reading is very low. It is mainly caused by the imbalanced labels in the training set since subjects usually experience much less time of having low oxygen level (e.g., below 90%) than having a normal oxygen level between 94% to 100%. Figure 12 shows another

example of the dynamics. As shown in the zoomed range, oxygen fluctuations correlate with one's sleep stage: in 'REM' (colored by light gray), the oxygen fluctuates drastically while in 'sleep' (colored by dark gray), the oxygen is much more stable and fluctuates in a small range. Our model makes accurate predictions since it leverages the sleep stage information.

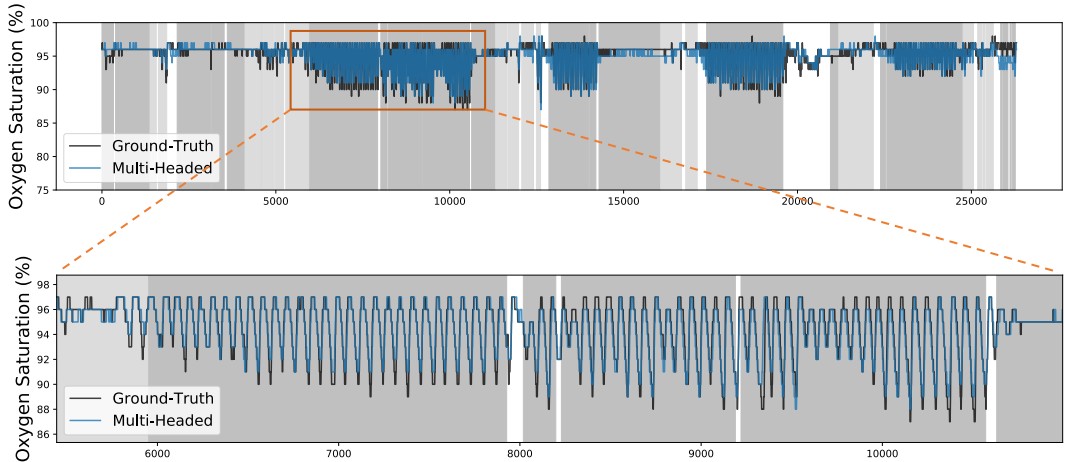

Figure 9: *MESA* Example 1.

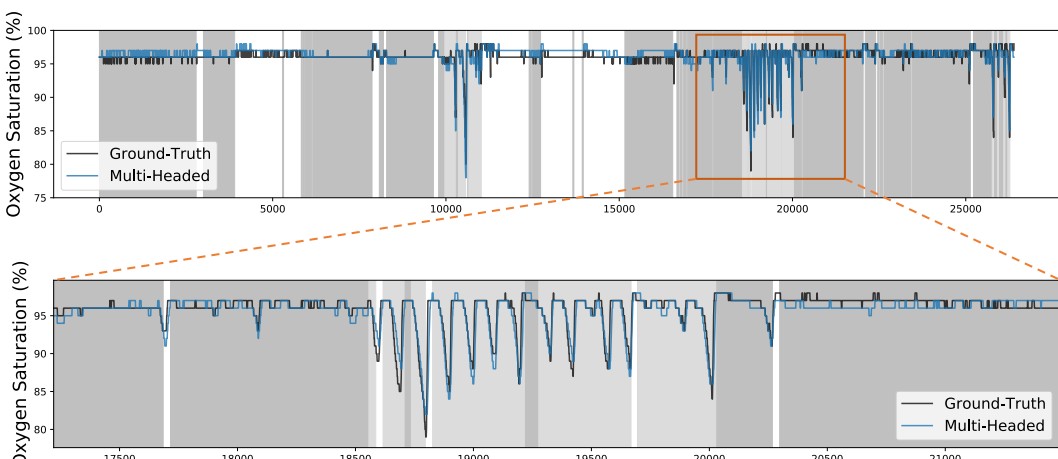

Figure 10: *MESA* Example 2.

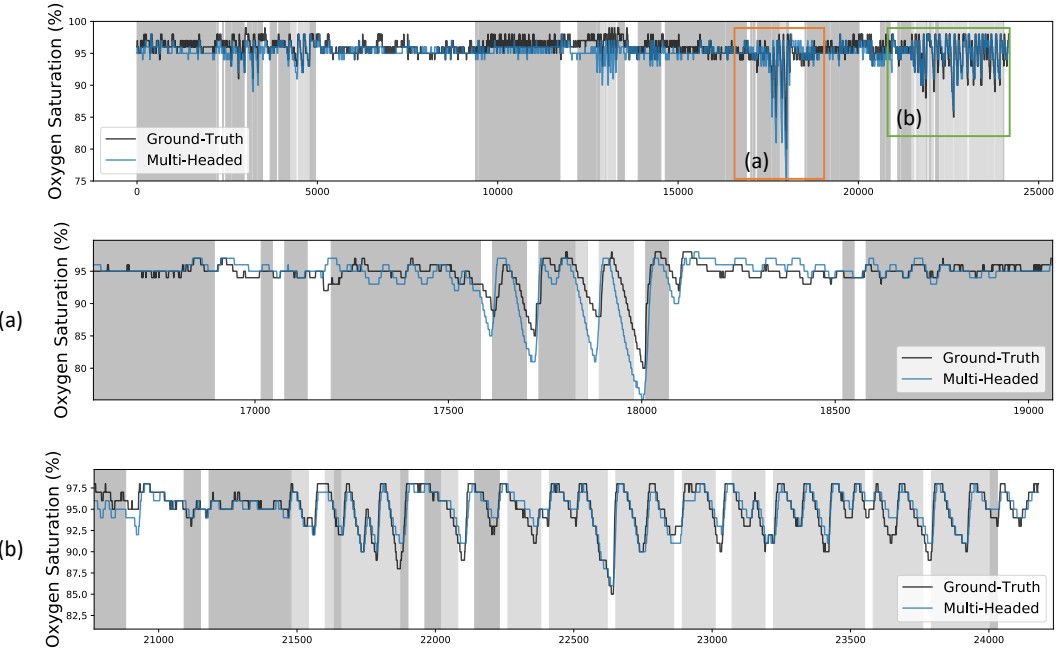

Figure 11: *MrOS* Example 1.

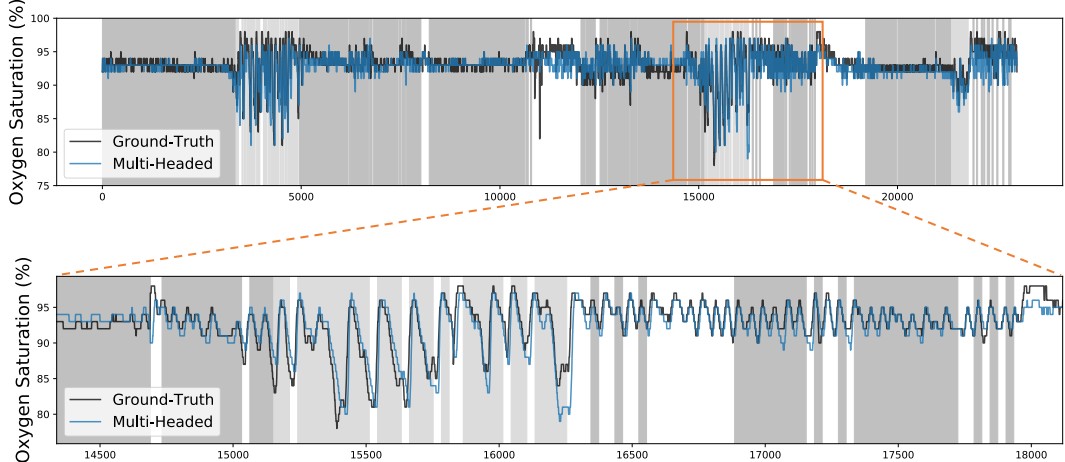

Figure 12: *MrOS* Example 2.

# D    COMPARISON AGAINST NAIVE BASELINE MODELS

We have compared our approaches with two naive baseline models, i.e., *Linear Regression* and *Random Forest Regression*. To choose features, we use the six statistical time series feature suggested by (Picard et al., 2001). Table 3 compares our model with the naive baselines. As shown, These baselines produce average prediction errors that are significantly higher than the multi-headed model. Figure 13 shows an illustrative example of the prediction. It can be noticed that both the linear regression and random forest models tend to make predictions concentrated at the averaged oxygen level without capturing the dynamics of oxygen fluctuations across time. While our model's prediction follows the ground truth oxygen changes well. It is because our model is designed to capture the non-local temporal dependencies in the signals.

Table 3: Prediction error (mean ± std) of naive baseline models compared with our multi-headed model.

| Model | SHHS | MESA | MrOS | Overall |
|---|---|---|---|---|
| Linear Regression | $1.86 \pm 1.72$ | $1.73 \pm 1.73$ | $1.87 \pm 1.69$ | $1.81 \pm 1.72$ |
| Random Forest | $1.85 \pm 1.73$ | $1.72 \pm 1.73$ | $1.89 \pm 1.71$ | $1.81 \pm 1.73$ |
| Multi-Headed | $\mathbf{1.62 \pm 1.54}$ | $\mathbf{1.48 \pm 1.51}$ | $\mathbf{1.66 \pm 1.51}$ | $\mathbf{1.58 \pm 1.49}$ |

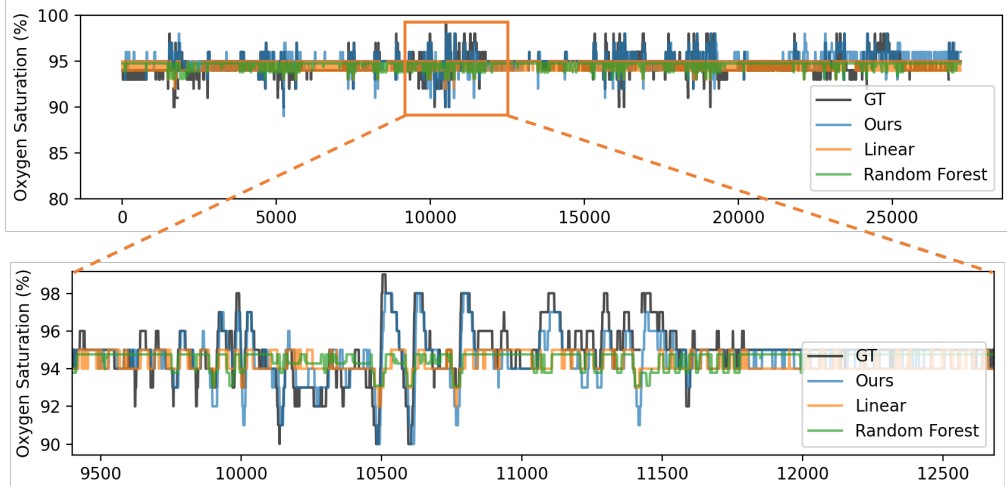

Figure 13: Comparison between our model and linear regression and random forest.

# E  SWITCHER-BASED AUXILIARY LEARNING OF A PIECE-WISE LINEAR FUNCTION

To highlight the benefits of switcher-based auxiliary learning and gradient diagnosis, we conduct experiments on the toy dataset shown in Figure 14. The target function is an 1-dimensional piece-wise linear function. The observed data has a small white noise added to the label. Visually the function looks like a string composed of 'W','N', and 'M'. Assume we have an auxiliary variable, *class*, that indicates which letter a data point belongs to. Our goal is to learn the function with access to the auxiliary variable. By using gradient diagnosis, it suggests that this auxiliary variable is a good "switcher" candidate. Denote $g_1$, $g_2$, $g_3$ as the gradients computed by the data points from the three letters. The cosine similarity between them are $\cos(g_1, g_2) = 0.48$, $\cos(g_1, g_3) = -0.02$, $\cos(g_2, g_3) = 0.39$, meaning the gradients are not aligned.

We compare our Multi-Headed model with three baselines: Single-Headed, Multi-Task, PCGrad-Adapted described in section 4. Figure 15 displays the learned function of the four methods. Only the Multi-Headed model successfully learns the target function and makes a prediction that visually looks like "WNM". Quantitatively, the mean squared errors of the predictions are 0.06, 0.06, 0.05, 0.01 for Single-Headed, Multi-Task, PCGrad-Adapted, and Multi-Headed, respectively. We attribute the Multi-Headed model's success to its usage of the auxiliary variable as a "switcher".

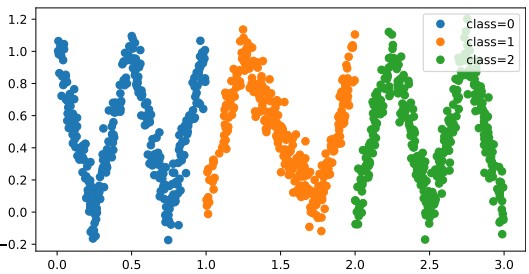

Figure 14: Visualization of the toy dataset. It is a 1D piecewise linear function. The color indicates the auxiliary variable for every data point.

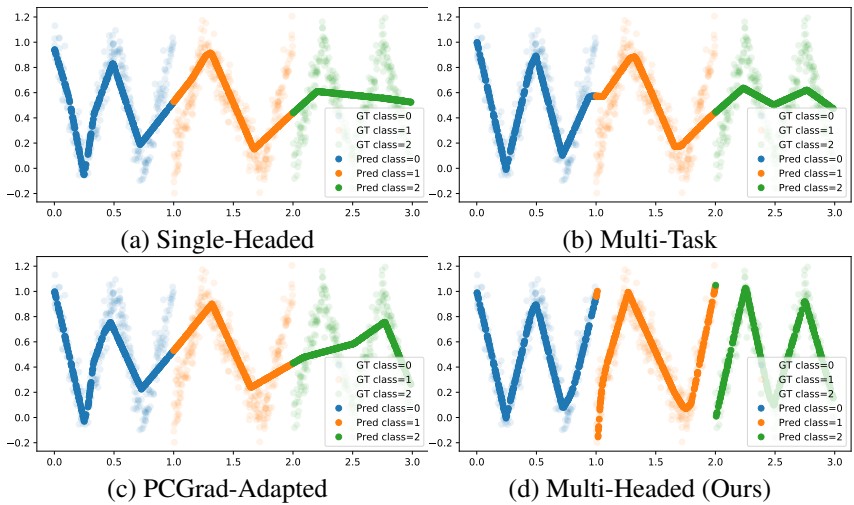

(a) Single-Headed      (b) Multi-Task

(c) PCGrad-Adapted      (d) Multi-Headed (Ours)

Figure 15: The benefits of switcher-based auxiliary learning. The figure shows the results of different models for the toy dataset. Only the multi-headed model which uses a switcher based method learns this non-smooth function effectively.

