# OpenReview forum: "Learning Blood Oxygen from Respiration Signals"
_ICLR.cc/2021/Conference — Reject_

### Official Review · AnonReviewer4 · 2020-10-29
**Nice model and experiments, but concerned about the prediction task.**

**Rating:** 3
**Confidence:** 4

**Review:**

Summary: A model that appears to very effectively predict SpO2 from respiration signals. I find the model and experiments well-designed, but have serious concerns about the motivation and prediction task itself.

**Update**: After reading the rebuttal and revised paper, I am keeping my score the same. There is a lot in this paper I really like -- an important prediction target and race analysis among them -- but still have concerns about clinical utility. In standard photoplethysmography, the causal graph is PaO2-->light absorbance, which leads to a clear story of what a fingertip SpO2 sensor is doing (modeling an imperfect but mostly unidirectional relationship). The causal graph in this paper is less clear but probably has some kind of cycle like ventilation-->SpO2-->ventilation, which is  a much more challenging story. I think there are two good routes this paper can take: (1) focus on how to clinically justify a model with such a potentially complicated causal graph; is the model learning the vent->O2 relationship, O2->vent, or some combination of the two? How can we be confident we know that? To which scenarios will and won't a model that has learned such relationships generalize? (The rebuttal has analysis that is a good start, but I think not enough to fully answer these questions) Or (2) acknowledge that the scope will necessary be limited when causal structure is so complicated and poorly understood, and actually limit the tasks the model is capable performing to those on which we can be confident it will do well. I think either (1) or (2) would involve major changes to the structure and goals of the paper, and probably require new experiments.

Objective: Predict blood oxygen sequence information from respiration sequence information, leverage auxiliary variables and demonstrate feasibility of RF-based contactless oxygen prediction. I view the first as the primary goal, bolstered by the auxiliary variables, which enables the RF-based prediction.

Strengths:
Overall, I think the model design and evaluation is very good! I like the paper a lot overall and think the authors did a good job addressing most methodological and experimental concerns.
* Impressive, accurate predictions.
* Experiments are well-designed. In particular, analysis w.r.t. race is well-motivated and well conducted.
* RF experiment demonstrates both RF feasibility and ability of the model to generalize across respiration measurement devices/methodologies.

Weaknesses:
Major weaknesses (only 1):
My most important concern is the choice of problem and prediction task. There is a well-discussed medical distinction between ventilation and respiration -- simply moving air vs actually adding oxygen to the blood stream. There is a good reason why respiratory rate and pulse oximetry are measured separately: resp. rate measures ventilation while oximetry measures respiration. Many conditions, i.e. pulmonary edema or embolism, affect the lungs' ability to exchange oxygen without affecting the ability to move air, and many conditions, i.e. traumatic or neurological, affect the ability to move air without affecting the ability to exchange oxygen in the lungs. Because of this, I see SpO2 as a variable that contains *independent* information not in the ventilation signal, and believe it is dangerous to display a patient SpO2 signal that is entirely *dependent* on ventilation - containing no respiration signal. Such a signal could look plausible enough but fail in the most important clinical cases.

I understand there is low error on the tasks shown, but how does the model work in cases where we'd expect respiratory problems but no ventilation problems -- or vice versa? Beyond sleep studies or ambulatory settings, is it likely to work in cases of pulmonary embolism or ARDS in an acutely ill COVID-19 patient? Some of the problems discussed in the paper (e.g., sleep-time drops in oxygen saturation) seem like they could plausibly be predicted from ventilation signal, but for these cases, why not predict a binary target instead of the full SpO2 signal? The only reason for predicting the SpO2 signal would seem to be if you think it's a true inference of the physiology that will hold true even in cases you didn't examine in the paper -- and I think there are a lot of prior physiological reasons to believe that's not the case. My prior is that there are at least 2 causal paths where you can predict SpO2 from ventilation signal: (1) hypoventilation-->hypoxemia and (2) hypoxemia-->hyperventilation. I'm concerned that the model may be able to pick up SpO2 signals that follow these patterns (and are accurate in the datasets used) without learning a general, "true" relationship between breathing and blood oxygen. This is important because the paper is primarily clinical: the value is the ability to predict a new clinical outcome. Thus, I think the paper should only be accepted if it gives a good idea of in what cases the model would be clinically useful at predicting SpO2, and I don't see much of this analysis in the paper. If these issues were discussed at all (and ideally in detail), I'd be more open to seeing clinical value in the work.

Minor points:
* Methodologically, it seems the auxiliary variable strategy works well but I'm not convinced it's the only way or the best way to solve the problem -- both multitask and multi-headed seem good at capturing the shape of the signal, with multitask often off by a constant. This is what we'd expect from an MSE model trained on a large dataset -- the overall signal shape will be pulled towards the mean. Gating into separate models for groups with different baselines would reduce this problem (because there will be less variance within each group). This is not a huge issue (and I know L1 loss is used here rather than MSE), but there are other ways to handle such problems -- for example using shape and time distortion losses like DILATE.
* I did not find the COVID analysis particularly compelling because there's only a single patient and mostly qualitative analysis -- it's hard to draw any clear conclusions form the example about the method's value overall in such cases. I think I'd prefer a more rigorous evaluation on a non-COVID task to something that's a bit speculative but COVID related.
* Some writing could be smoother and more terse in the Method section, i.e. " In this paragraph, we answer
the critical question..."

---

> ### Author Response · Authors · 2020-11-25
> **Response to Reviewer 4 (3/3)**
>
> **Q2: Using the DILATE loss.**
>
> Thanks for the suggestion. The original DILATE paper is not directly applicable to our problem. The DILATE paper focuses on multi-step forecasting problems for non-stationary signals. The number of steps K is small (K < 100) because the computation complexity of DILATE grows quickly with K. We can’t apply DILATE directly to our task since our task is not a forecasting problem, and the length of the predicted oxygen is huge, i.e., it can be more than 50,000 steps.  We may adapt DILATE to our model by applying the DILATE loss on segments, where each segment is a sequence of many steps (e.g., 100 steps). However, the choice of loss is orthogonal to the choice of model. We could use both the DILATE loss and the multi-headed model to improve the performance.
>
> **Q3: Only 1 Covid patient is presented and most analysis is qualitative.**
>
> The reference to COVID was mainly for motivation; we only meant to say that if there is a solution that allows monitoring the oxygen of COVID patients from a distance and without body contact  (i.e., via radio signals), it would be beneficial. We have modified the abstract and introduction to tune down the reference to COVID.

---

> ### Author Response · Authors · 2020-11-25
> **Response to Reviewer 4 (2/3)**
>
> **(3) Is it OK to use the breathing signal to predict oxygen level?** It is important to clarify that we do not use the breathing rate to estimate oxygen. We rather use the detailed breathing signal which has significantly more information about oxygen saturation than breathing rate, as indicated by Figure 8 in Appendix B, which we discussed above.
>
> We agree with the reviewer that breathing and oxygen saturation are different metrics but they are related through chemoreflexes [12]. In the medical domain, it is common to use indirect or surrogate metrics when the target measurement is invasive or impractical. In fact, oximetry itself is a surrogate. The real metric of interest is the fraction of oxygen-saturated hemoglobin relative to total hemoglobin in arterial blood, or so-called SaO2. A direct measurement of SaO2 requires blood samples, and hence is invasive. Pulse oximetry provides a non-invasive surrogate of SaO2. Instead of measuring arterial blood oxygen, oximetry measures the relative absorbance of red light through a finger. Since this is a surrogate, it is surely possible that there exist errors, or discrepancies with respect to the real target. The medical literature reports that, for COVID‐19 patients in a critical care unit, oxygen saturation measured by pulse oximetry was consistently and significantly lower than arterial oxygen saturation measured directly by blood gas analysis [8]. These discrepancies are not limited to COVID-19. The literature also reports that in the critically ill, oximetry does not reliably predict changes in SaO2  because the calibration of red light absorbance is based on calculations made from healthy volunteers [7].  Indeed, oximetry fails in many important scenarios including subjects having chronic obstructive pulmonary disease (e.g., chronic bronchitis and emphysema) [1],  anemia [2], sickle cell disease [3], or skin pigmentation [4].
>
> Despite that oximetry does not work in some diseases and health conditions, it remains a key surrogate for arterial oxygen saturation. In this paper, we aim to provide another surrogate based on breathing signals. The objective is to provide a method that is non-invasive, easy to use, non-contact, and can be useful in many cases, particularly for old people, who are naturally at high risk of low oxygen [10] and are well represented in our evaluation datasets.   Furthermore, old patients can have onset of dementia or Alzheimer’s and cannot remember to regularly measure themselves using a pulse oximeter. Thus, they would benefit from a system that passively and continuously measures oxygen level using radio signals without asking the patient to remember to actively measure themselves.
>
> **Reference**
> [1] Amalakanti S, Pentakota M R. Pulse oximetry overestimates oxygen saturation in COPD[J]. Respiratory Care, 2016.
> [2] Severinghaus J W, Koh S O. Effect of anemia on pulse oximeter accuracy at low saturation[J]. Journal of clinical monitoring, 1990.
> [3] Blaisdell C J, Goodman S, Clark K, et al. Pulse oximetry is a poor predictor of hypoxemia in stable children with sickle cell disease[J]. Archives of pediatrics & adolescent medicine, 2000.
> [4] Bickler P E, Feiner J R, Severinghaus J. W. Effects of skin pigmentation on pulse oximeter accuracy at low saturation[J]. Anesthesiology: The Journal of the American Society of Anesthesiologists, 2005.
> [5] Barker S J, Hyatt J, Shah N K, et al. The effect of sensor malpositioning on pulse oximeter accuracy during hypoxemia[J]. Anesthesiology: The Journal of the American Society of Anesthesiologists, 1993.
> [6] Sverrir I. Gunnarsson  Paul E. Peppard  Claudia E. Korcarz  Jodi H. Barnet  Erika W. Hagen  K. Mae Hla  Mari Palta  Terry Young  James H. Stein. Minimal nocturnal oxygen saturation predicts future subclinical carotid atherosclerosis: the Wisconsin sleep cohort. Journal of Sleep Research, Volume 24, Issue 6, June 2015.
> [7] Perkins GD, McAuley DF, Giles S, Routledge H, Gao F. Do changes in pulse oximeter oxygen saturation predict equivalent changes in arterial oxygen saturation? Critical Care 2003.
> [8] N. Wilson‐Baig, T. McDonnell,  and A. Bentley. Discrepancy between SpO2 and SaO2 in patients with COVID‐19. Journal of Anaesthesia, August 2020.
> [9] G. Iyer Parameswaran, Kathy Brand, James Dolan. Pulse Oximetry as a Potential Screening Tool for Lower Extremity Arterial Disease in Asymptomatic Patients With Diabetes Mellitus. JAMA Internal Medicine, February 2005.
> [10] NLM. Aging changes in the lungs. https://medlineplus.gov/ency/article/004011.htm, 2020.
> [11] Luks A M, Swenson E R. Pulse Oximetry for Monitoring Patients with COVID-19 at Home: Potential Pitfalls and Practical Guidance[J]. Annals of the American Thoracic Society, 2020 (ja).
> [12] Maria Plataki, Scott A. Sands, and Atul Malhotrab. Clinical consequences of altered chemoreflex control. Respiratory Physiology & Neurobiology Journal. November, 2013.

---

> ### Author Response · Authors · 2020-11-25
> **Response to Reviewer 4 (1/3)**
>
> We thank the reviewer for the insightful comments. We are glad the reviewer finds the model design and evaluation to be very good and likes the paper overall. The reviewer also raises important and deep clinical issues that we address below.
>
> **Q1: The reviewer is concerned about the choice of problem. In particular, the reviewer asks about the clinical usefulness of the model with patients, and whether the model is only focused on ventilation.**
>
> Below we respond to the reviewer’s concerns.  Our response has three parts. We first show that our model performs well in realistic clinical settings across many diseases, including diseases that affect oxygen saturation, and where the mechanism of impacting oxygen is unrelated to ventilation. Second, we use empirical results to argue that the model learns complex relationships between breathings signals and oxygen that cannot be captured by simply looking at breathing rate.  Third, we argue that while the model may not apply in all situations, it is clinically useful for monitoring the elderlies who are particularly at a high risk of low oxygen, and who are highly represented in our evaluation datasets.
>
> **(1) The model performs well on patients including patients whose diseases interact with oxygen and the mechanism of interaction is unrelated to ventilation.** Our evaluation datasets include patients with a variety of diseases. In particular, the Multi-Ethnic Study of Atherosclerosis (MESA) dataset focuses on patients with Atherosclerosis, a disease in which cholesterol plaques build up in the walls of arteries, and the minimum oxygen saturation predicts future disease prognosis [6]. Clearly, the interaction between this disease and oxygen saturation is not related to ventilation (i.e., air in and out). Yet, as shown in the paper, our model has low error and good performance on this dataset.  MESA is not the only dataset with patients who suffer from diseases that interact with oxygen saturation. Our three evaluation datasets (SHHS, MrOS and MESA) contain people who have various diseases and health conditions. This includes chronic bronchitis (323 subjects), cardiovascular diseases (1196 subjects), coronary heart disease (869 subjects), myocardial infarctions (508 subjects), diabetes (307 subjects), Alzheimer’s (40 subjects), Parkinson’s (69 subjects), and others. This means that our reported accuracy already takes into account unhealthy individuals. Please note that many of these diseases, including diabetes, cardiovascular diseases, and atherosclerosis, interact with oxygen saturation through mechanisms other than ventilation.  Our model performs well on this population, as shown in our evaluation results.  Furthermore, all three datasets are focused on old people. In particular, the median age in MrOS is 80 years. Hence, 50% of the subjects are in their 80’s and 90’s. Such old subjects are natural beneficiaries from monitoring oxygen saturation since they are at a high risk of low oxygen [10].  Our model performs very well on those subjects.
>
> **(2) The model learns complex relationships between breathing signals and oxygen that cannot be captured by simply looking at breathing rate and instantaneous ventilation:**  It is important to note that we do not use the breathing rate to estimate oxygen. We rather use the detailed breathing signal which has significantly more information about oxygen saturation than the breathing rate. The model leverages the breathing signal to extract information far beyond the breathing rate.  To see this, please consider Figure 8 in Appendix B, which shows 4 different cases. In each of these cases, there are two patients who have the same breathing rate, yet their oxygen level is very different. The figure shows that our model estimates the oxygen level for each of those subjects correctly, and does not predict the same oxygen level even though the breathing rate is the same. Based on Figure 8 and the signals therein, it is clear that the model must be looking into the patterns and shapes of the signals, not just the breathing rate, and likely is leveraging the temporal dependencies (using its Transformer-based architecture), which allows it to discover complex relationships between breathing and oxygen by looking holistically at the breathing signal.
>
> We note that, physiologically, oxygen affects breathing through the peripheral chemoreceptors, which are located in the carotid bodies, and primarily respond to changes in blood oxygen  [12]. There is a feedback loop that relates blood oxygen and ventilation, allowing changes in oxygen to cause changes in breathing.  However, the details of the chemoreflex physiology are complex and not yet fully understood [12]. Our approach is data-driven and demonstrates that custom neural networks can extract significant and useful information about oxygen levels from the breathing signal.
>
> **(See point 3 for Q1 in the next post.)**

---

### Official Review · AnonReviewer1 · 2020-10-29
**Encouraging result in a very topical domain**

**Rating:** 6
**Confidence:** 4

**Review:**


##########################################################################

Summary:


The paper proposes neural network models to predict oxygen saturation from breathing signals. The architecture implements "feature switches" that partition the data in a multi-head manner with the ultimate goal of predicting auxiliary variables which will help the prediction. The results are very encouraging especially given the immediate application to COVID-19 non-invasive monitoring.

##########################################################################

Reasons for score:


Overall, I vote for marginal accept. I like the idea of predicting one modality from another. My major concern is about the experimental section of the paper and some additional ablations (see cons below). Hopefully, the authors can address my concern in the rebuttal.


##########################################################################Pros:


1. The paper leverages one of the most important biomarkers for COVID-19 monitoring. If it's possible to predict SpO2 robustly from other signals, it's a big advance.

2. The proposed architecture seems to be superior to vanilla models, as reported in the experiments section.


3. This paper provides comprehensive experiments, including both qualitative analysis and quantitative results, to show the effectiveness of the proposed framework.


##########################################################################

Cons:


1. My major concern is the use of PSG and RF signals. Since both of them require prohibitively-expensive equipment (a minimum of 22 wire attachments, and a big antenna I suppose, respectively), I am curious why the paper is not using easier to obtain sensors. For example, some of the medical datasets already used here include heart rate measured through PPG signals or ECG (now common in smartwatches). Could the same framework be applied to heart rate->SpO2?


2. The motivation behind the feature switches seems a bit inflated. The paper ends up using the gradient cosine similarities to inform the final outputs. Why do we need to inspect the gradients in contrast to -say- the validation loss? Couldn't this be achieved with traditional feature importance methods? For example, permutation feature importance could inform which feature set is insignificant or you could even apply SHAP to the vanilla model with all features.

3. In terms of baselines, the paper starts from a single-headed neural network. However, it would be good to explore the "lower bound" of the dataset with a naive model. What would be the error of linear regression or a random forest regressor trained on features extracted by the signals? I am aware that here the output is multi-step, but you could apply the models iteratively on the predicted values. Also, in general, I would like to see more information about the parameter size of the models, batch size etc. In the appendix, we read that "the breathing signals we used in all the experiments last several hours per night", which means that the output is also a signal that spans hours? How do you average the error for that kind of multi-step forecasting? Does the error increase as we forecast further in the future?


##########################################################################

Questions during the rebuttal period:


Please acknowledge the potential use of cheaper and more accessible sensors.

The structure of the sections could be improved as well. Section 3 is not the right place to mention (some) results.

Figure 4 --> is this an observation made for the Multi-head model only? How is the distribution of the single-headed model?

learning to throw the data to the right function --> informal, please rephrase



#########################################################################

Many typos (please proofread the entire paper again):

-These result is quite interesting

-there are medical evidence

-..

---

> ### Author Response · Authors · 2020-11-25
> **Response to Reviewer 1 (3/3)**
>
> **Q3. Adding naive baselines like linear regression and random forest.**
>
> We thank the reviewer for the suggestion. We have run the experiments for both linear regression and random forest on the three medical datasets. These baselines produce average prediction errors that are 14% to 17% higher than our multi-headed model. In our revision, we include a comparison between our model and those baselines in Appendix D. Table 3 lists the error and Figure 13 shows an illustrative example of the prediction. It can be noticed that both linear regression and random forest tend to make a prediction concentrated at the averaged oxygen level without capturing the dynamics of oxygen fluctuation across time. While our model’s prediction follows the ground truth oxygen changes well. It is because our model is designed to capture the non-local temporal dependencies in the signals.
>
> **Q4. Are we doing a multi-step forecasting problem?**
>
> Our task is NOT a multi-step forecasting task. As described in section 3.1, our task is predicting oxygen from respiration signals, as opposed to forecasting the oxygen in the future. Mathematically, our model predicts oxygen $y_i$ from breathing $x$ by $p(y_1, y_2, \cdots, y_T | x_1, x_2, \cdots, x_T)$ instead of $p(y_i | x_1, x_2, \cdots, x_i, y_1, y_2, \cdots, y_{i-1})$. The model’s output does not depend on its previous predictions. Therefore, there is no error accumulation along the time axis. We have updated the text to make it clearer.
>
> **Q5. Is the observation of figure 4 also held for the Single-Headed model?**
>
> The observation in Figure 4 is related to showing that our model’s prediction has no bias against different races. This property stems from the fact that our model uses breathing signals to predict oxygenation as opposed to sensing the “red color“ of blood through the skin, as is the case for pulse oximetry. Since breathing is not biased by skin color, our approach eliminates the root cause for such bias. Therefore, the observation applies to all models proposed in the paper. Please note that we are the first to show that one can get oxygen saturation from breathing. The multi-headed model and the single-headed model are both contributions of this paper.
>
> **Q6. Other issues about presentation.**
> We thank the reviewer for the suggestions. We have revised and reorganized the paper as suggested.
> (1) The number of parameters for the multi-headed model is 26,821,113 and the model size is 107.28MB. During training, we set the batch size for all models to 1 (i.e., one night of signal which includes tens of thousands of oxygen measurements) due to a varying input length. The details of all models and the training parameters are included in Appendix A.
> (2)  We moved the gradient diagnosis results from section 3.3 to section 4.3.
> (3)  We replaced the sentence “learning to throw the data to the right function” in section 3.3 with a more formal description.

---

> ### Author Response · Authors · 2020-11-25
> **Response to Reviewer 1 (2/3)**
>
> **Q2. The motivation of the “feature switch” seems a bit inflated … Why inspect gradients instead of using validation loss or feature importance methods?**
>
> Inspecting gradients is much more efficient compared with checking the validation loss. Computing the validation loss requires training a new model for every auxiliary variable. While our gradient diagnosis only requires computing the gradient of one trained model which is several orders of magnitude faster.
>
> As for feature importance methods, including random forest-based importance [3], permutation importance [2], or SHAP [1], they only provide the importance of the variable as an input feature. However, a variable’s inability of improving the model’s prediction as features does not mean the variable is useless. For example, in our task, using sleep stages as an input feature empirically does not improve performance. However, using it as a switcher in a multi-headed model does improve performance. Our gradient diagnosis evaluates a variable’s “importance” as “switcher”, which is a novel way of evaluating and using auxiliary variables.
>
> We also note that the idea of using auxiliary variables as switchers is novel and significantly different from existing solutions. Current solutions use auxiliary variables either as input or as an auxiliary output task. This however assumes that the relationship between the auxiliary variable and the learned function is smooth. In contrast, our switcher-based model does not assume the relationship to be necessarily smooth or continuous since it allows the network to learn different manifolds depending on the value of the auxiliary variable and switch between them. Our gradient test is crafted to check whether an auxiliary variable is better used as a switcher. We have added a toy task in Appendix E of the revised paper to highlight this aspect of the model.
>
> **Reference**
> [1] Lundberg, Scott M., and Su-In Lee. "A unified approach to interpreting model predictions." Advances in neural information processing systems. 2017.
> [2] Altmann, André, et al. "Permutation importance: a corrected feature importance measure." Bioinformatics 26.10 (2010): 1340-1347.
> [3] Breiman, Leo, et al. "Classification and regression trees. Statistics/probability series." (1984).

---

> ### Author Response · Authors · 2020-11-25
> **Response to Reviewer 1 (1/3)**
>
> We thank the reviewer for the constructive comments, and respond below to the points raised in the review.
>
> **Q1. PSG and RF signals seem hard to acquire. Why not use heart rate measured through PPG signals or ECG (now common in smartwatches). Could the same framework be applied to heart rate->SpO2?**
>
> Below, we clarify that obtaining breathing signals is relatively easy; we then extend the model to predict SpO2 from heart rate and show that the resulting error rate is significantly higher than predicting SpO2 from breathing.
>
> Past work has shown that accurate breathing signals can be easily obtained from standard RF devices. References at the end of this section show that one can obtain breathing signals from commodity WiFi routers [1,2,3], wideband radios [4,5], and even RFIDs [6]. One can also get breathing signals using a microphone and a smartphone app [7,8].
>
> As for PSG, it is important to note that we do not use the full PSG data.  We only use breathing signals, which are obtained using a wearable belt that stretches with the inhale-exhale motion [9]. Please note that since we do not use the EEG data in PSG, there is no need to wear 22 wires or electrodes. Also please note that while we use sleep stages, we do not take them from PSG, we actually predict them from the breathing signal.
>
> We also followed the reviewer’s suggestion of estimating SpO2 from heart rate since as noted by the reviewer it is easy to obtain heart rate using smartwatches. Our framework can be directly adapted to predict SpO2 from heart rate. Since the PSG data includes heart rate, we used it to train the model to predict SpO2 from heart rate. However, the error is  1.91 in SHHS, which is significantly higher than our model (1.62).
>
> **Reference**
> [1] Wang, Xuyu, Chao Yang, and Shiwen Mao. "TensorBeat: Tensor decomposition for monitoring multi person breathing beats with commodity WiFi." ACM Transactions on Intelligent Systems and Technology (TIST) 9.1 (2017): 1-27.
> [2] Wang, Xuyu, Chao Yang, and Shiwen Mao. "PhaseBeat: Exploiting CSI phase data for vital sign monitoring with commodity WiFi devices." 2017 IEEE 37th International Conference on Distributed Computing Systems (ICDCS). IEEE, 2017.
> [3] H. Abdelnasser, K. A. Harras, and M. Youssef. 2015. Ubibreathe: A ubiquitous non-invasive WiFi-based breathing estimator. In Proceedings of the IEEE MobiHoc Conference (MobiHoc’15). ACM, New York, NY, 277--286
> [4] Yang, Yanni, et al. "Multi-breath: Separate respiration monitoring for multiple persons with UWB radar." 2019 IEEE 43rd Annual Computer Software and Applications Conference (COMPSAC). Vol. 1. IEEE, 2019.
> [5] Shichao Yue et al. "Extracting multi-person respiration from entangled RF signals." Proceedings of the ACM on Interactive, Mobile, Wearable and Ubiquitous Technologies 2.2 (2018): 1-22.
> [6] Hou, Yuxiao, Yanwen Wang, and Yuanqing Zheng. "Tagbreathe: Monitor breathing with commodity rfid systems." 2017 IEEE 37th International Conference on Distributed Computing Systems (ICDCS). IEEE, 2017.
> [7] Wang, Xuyu, Runze Huang, and Shiwen Mao. "SonarBeat: Sonar phase for breathing beat monitoring with smartphones." 2017 26th International Conference on Computer Communication and Networks (ICCCN). IEEE, 2017.
> [8] Rajalakshmi Nandakumar, Shyam Gollakota, and Nathaniel Watson. Contactless Sleep Apnea Detection on Smartphones, ACM MobiSys, 2015.
> [9] Respironics breathing belt. https://www.medexsupply.com/diagnostic-tools-diagnostic-stations-and-accessories-pro-tech-zrip-dura-belt-x_pid-105444.html?pid=105444

---

### Official Review · AnonReviewer2 · 2020-10-30
**An ok paper with a potentially misleading abstract**

**Rating:** 4
**Confidence:** 4

**Review:**

Overview: The authors present an approach to learn blood oxygen levels using respiratory signal data. This might be relevant for various contexts such as ARDS or COVID-19. Unlike using pulse oximetry, it doesnt rely on wearing a sensor and less biased on darker skinned individuals.

Clarity and Quality: The problem description and aim of the paper are very clear. Unfortunately, I think the abstract of the paper and the introduction are quite misleading however as the authors entirely focus on COVID-19, while only a tiny fraction of the results are actually demonstrated on COVID. The paper thus makes bigger claims for COVID than what is demonstrated in the experiments which is misleading.

Originality and Significance: Certainly the problem of learning blood oxygen levels is a significant one in high-risk patients in ICU for instance. However, the originality of the modelling approach is not that unique. Specifically the authors use auxiliary learning to improve on standard ML approaches. The model proposed is a multiheaded model that incorporates auxiliary oxygen-related variables into the prediction process.

Pros: 1. The paper addresses a relevant biomedical problem
2.  The authors demonstrate the performance of their approach on three separate datasets

Cons:
1. Unfortunately, the authors never explicitly clarify what is meant by a "breathing signal". Is this the respiratory rate or a pulse or something else? I find this term neither technically sound from the medical perspective nor informative to a non-expert. This is also crucial to understanding how good the model is. For instance, if the task is to predict blood oxygen from "breathing signals", how noisy are these measurements? How are these recorded? Are these recorded via contactless sensing?

2. In the experiment for COVID, the model is only applied to a case of a single COVID-19 patient. The reader needs more information of what is the prior condition of the patient, is he in ICU? If so, how do you account for treatment policies or protocol changing for the duration of stay; this could affect overall outcomes (and oxygen saturation)? Moreover, the paper oversells this demonstration on a single COVID-19 patient almost as its key contribution. Yet this is only a single patient.

---

> ### Author Response · Authors · 2020-11-25
> **Response to Reviewer 2 (2/2)**
>
> **Q3. The originality of the approach is not unique.**
>
> We would like to highlight that while auxiliary learning is a known method, to the best of our knowledge, we are the first to propose using auxiliary variables as “switchers” in a multi-headed model. This is a significant difference from existing solutions, which use auxiliary variables either as input or as an auxiliary output task. Such models implicitly assume that the relationship between the auxiliary variable and the learned function is smooth. In contrast, our model is suitable when the relationship is not necessarily smooth or continuous since it allows the network to learn different manifolds depending on the value of the auxiliary variable. Our gradient test is crafted to check whether an auxiliary variable is better used as a switcher. We have added a toy task in Appendix E of the revised paper to highlight this aspect of the model.
>
> Beyond the introduction of switcher-based auxiliary learning, the model demonstrates the benefits of using Transformers to capture non-local dependencies in physiological signals. This is in contrast to previous ML models for processing physiological data such as breathing [1], ECG [2], EEG [3], which use CNN and RNN. We note however that applying Transformers to physiological data is non-trivial since they are large time series of over 10,000 time steps. To handle this issue, we employ an encoder-decoder structure and apply the transformer in the bottleneck layer.
>
> **Reference**
> [1] Zhao, Mingmin, et al. "Learning sleep stages from radio signals: A conditional adversarial architecture." International Conference on Machine Learning. 2017.
> [2] Kiyasseh, Dani, Tingting Zhu, and David A. Clifton. "CLOCS: Contrastive Learning of Cardiac Signals." arXiv preprint arXiv:2005.13249 (2020).
> [3] Roy, Yannick, et al. "Deep learning-based electroencephalography analysis: a systematic review." Journal of neural engineering 16.5 (2019): 051001.

---

> ### Author Response · Authors · 2020-11-25
> **Response to Reviewer 2 (1/2)**
>
> We thank the reviewer for acknowledging the significance of our problem, learning blood oxygen levels. We address your concerns as follows.
>
> **Q1. The COVID part of abstract and introduction is misleading. The model is only applied to a case of a single COVID-19 patient.**
>
> The reference to COVID was for motivation; we only meant to say that if there is a solution that allows monitoring the oxygen of COVID patients from a distance and without body contact  (i.e., via radio signals), it would be beneficial. We agree with the reviewer that the reference to COVID was over-emphasized. We have modified the abstract and introduction to tune down the reference to COVID and be clear that it is only for motivation and not a contribution.
>
> We would like to clarify that the contributions of this paper are:  1) introducing the new task of predicting oxygen level from breathing signals, 2) developing and evaluating a deep learning solution for this task, and 3) demonstrating on three medical datasets and one RF dataset that our solution estimates oxygen level accurately, and works without physical contact with the patient’s body because it can measure breathing using radio signals.
>
> **Q2. “Breathing signal” is not explicitly clarified.**
>
> We have updated the text in the paper to provide a definition of the breathing signal and an illustrative figure. In particular, the input to our model is the breathing signal, which is a time series that measures the chest displacement that follows the inhale-exhale motion. A visualization of breathing signals is presented in Figure 7 of the appendix. The breathing signal can be obtained using wearable devices or contactless sensors. For example, the person can wear a breathing belt that measures the breathing signal from the amount of stretching that occurs due to the inhale-exhale motion.  Alternatively, the breathing signal can be measured without body contact. For example, references [1,2,3,4] below use radio signals (e.g., Wifi) to measure the breathing signal without body contact, whereas [5] measures breathing signals using the microphone on a smartphone. We showed empirically in the paper that our model works whether the breathing signal is measured using a breathing belt or radio signals.
>
> **Reference**
> [1] Adib, Fadel, et al. "Smart homes that monitor breathing and heart rate." Proceedings of the 33rd annual ACM conference on human factors in computing systems. 2015.
> [2] Yue, Shichao, et al. "Extracting multi-person respiration from entangled RF signals." Proceedings of the ACM on Interactive, Mobile, Wearable and Ubiquitous Technologies 2.2 (2018): 1-22.
> [3] H. Abdelnasser, K. A. Harras, and M. Youssef. 2015. Ubibreathe: A ubiquitous non-invasive WiFi-based breathing estimator. In Proceedings of the IEEE MobiHoc Conference (MobiHoc’15). ACM, New York, NY, 277--286
> [4]  X. Wang, C. Yang, S. Mao, TensorBeat: Tensor Decomposition for Monitoring Multi-person Breathing Beats with Commodity WiFi, ACM Transactions on Intelligent Systems and Technology, September 2017.
> [5] Rajalakshmi Nandakumar, Shyam Gollakota, and Nathaniel Watson. Contactless Sleep Apnea Detection on Smartphones, ACM MobiSys, 2015.

---

### Author Response · Authors · 2020-11-25
**Revision of Manuscript**

We thank all the reviewers for providing us with valuable feedback. We have modified the main paper as well as the appendix, as per comments. Hope the revision could address the concerns from the reviewers. Here are the major changes that we made to the manuscripts. All modifications are highlighted by the red color in the revision.

In the Appendix:
1. We have modified **Appendix A** to incorporate more training details.
2. We have added **Appendix B** to clarify and visualize the breathing signals, discuss the difference between breathing signals and breathing rate, and introduce the measurement of breathing signals.
3. We have added **Appendix D** to compare our method with naive baseline models quantitatively and qualitatively.
4. We have added **Appendix E** to provide a toy example to illustrate the effectiveness of our multi-headed model and gradient diagnosis.

In the main paper:
1. We have modified the **abstract** and **introduction** to tune down the reference to COVID-19.
2. We have modified **Section 3** to make the problem formulation and the definition of breathing signal more clear.
3. We have improved the structure by moving the gradient diagnosis results **from Section 3 to Section 4**.
4. We have added **Section 5** to discuss the clinical usage of our method.

---

### Decision · Program_Chairs · 2021-01-07
**Final Decision**

**Decision:**

Reject

**Comment:**

Two referees indicate reject, one supports (weak) accept. My impression is that major points of criticism raised by the reviewers -- mostly about limited novelty and somewhat inconclusive experimental results -- could not be addressed in a clearly convincing way during the rebuttal phase. I will therefore recommend rejection.